# Percutaneous Coronary Intervention (PCI) Reprograms Circulating Extracellular Vesicles from ACS Patients Impairing Their Cardio-Protective Properties

**DOI:** 10.3390/ijms221910270

**Published:** 2021-09-24

**Authors:** Saveria Femminò, Fabrizio D’Ascenzo, Francesco Ravera, Stefano Comità, Filippo Angelini, Andrea Caccioppo, Luca Franchin, Alberto Grosso, Cecilia Thairi, Emilio Venturelli, Claudia Cavallari, Claudia Penna, Gaetano Maria De Ferrari, Giovanni Camussi, Pasquale Pagliaro, Maria Felice Brizzi

**Affiliations:** 1Department of Medical Sciences, University of Turin, 10126 Turin, Italy; saveria.femmino@unito.it (S.F.); Francesco.ravera@edu.unito.it (F.R.); andrea.caccioppo@gmail.com (A.C.); alberto.grosso@edu.unito.it (A.G.); emilio.venturelli@edu.unito.it (E.V.); giovanni.camussi@unito.it (G.C.); 2Department of Medical Sciences, Division of Cardiology, University of Turin, 10126 Turin, Italy; fabrizio.dascenzo@unito.it (F.D.); filippoangelini90@gmail.com (F.A.); luca.franchin@gmail.com (L.F.); gaetanomaria.deferrari@unito.it (G.M.D.F.); 3Department of Clinical and Biological Sciences, University of Turin, 10143 Orbassano, Italy; stefano.comita@unito.it (S.C.); cecilia.thairi@unito.it (C.T.); claudia.penna@unito.it (C.P.); pasquale.pagliaro@unito.it (P.P.); 42i3T Scarl, University of Turin, 10126 Turin, Italy; cavallari.1184@gmail.com

**Keywords:** PCI, extracellular vesicles, ACS, ischemia/reperfusion injury

## Abstract

Extracellular vesicles (EVs) are promising therapeutic tools in the treatment of cardiovascular disorders. We have recently shown that EVs from patients with Acute Coronary Syndrome (ACS) undergoing sham pre-conditioning, before percutaneous coronary intervention (PCI) were cardio-protective, while EVs from patients experiencing remote ischemic pre-conditioning (RIPC) failed to induce protection against ischemia/reperfusion Injury (IRI). No data on EVs from ACS patients recovered after PCI are currently available. Therefore, we herein investigated the cardio-protective properties of EVs, collected after PCI from the same patients. EVs recovered from 30 patients randomly assigned (1:1) to RIPC (EV-RIPC) or sham procedures (EV-naive) (NCT02195726) were characterized by TEM, FACS and Western blot analysis and evaluated for their mRNA content. The impact of EVs on hypoxia/reoxygenation damage and IRI, as well as the cardio-protective signaling pathways, were investigated in vitro (HMEC-1 + H9c2 co-culture) and ex vivo (isolated rat heart). Both EV-naive and EV-RIPC failed to drive cardio-protection both in vitro and ex vivo. Consistently, EV treatment failed to activate the canonical cardio-protective pathways. Specifically, PCI reduced the EV-naive Dusp6 mRNA content, found to be crucial for their cardio-protective action, and upregulated some stress- and cell-cycle-related genes in EV-RIPC. We provide the first evidence that in ACS patients, PCI reprograms the EV cargo, impairing EV-naive cardio-protective properties without improving EV-RIPC functional capability.

## 1. Introduction

Acute Coronary Syndrome (ACS) refers to a spectrum of clinical presentations of ischemic heart disease (IHD) comprising unstable angina, non-ST-elevated myocardial infraction (NSTEMI) and ST-elevated myocardial infraction and represents one of the major causes of death worldwide. The revascularization approach, particularly percutaneous coronary intervention (PCI), is the backbone treatment option for ACS [1]. Indeed, reperfusion is vital to avoid the evolution of myocardial damage and to improve patients’ outcomes. However, reperfusion per se can induce damage through a phenomenon denoted as ischemia/reperfusion injury (IRI), which may offset the benefits of percutaneous revascularization [2]. IRI is the result of the activation of several damaging pathways, including oxidative stress [3,4], the increase in intracellular calcium [5], the restoration of the physiological pH [6] and the inflammatory process [7]. This may translate into discrete functional entities such as myocardial stunning, the no-reflow phenomenon, reperfusion arrhythmias and lethal reperfusion injury [2]. Among these, the lethal reperfusion injury represents an additional inducer of cell death, distinct from the ischemic damage [8]. Therefore, the identification of new potential IRI targets is crucial to improve the benefits associated with the current treatment options in patients suffering from ACS.

Cardio-protection refers to all feasible interventions that aim to attenuate myocardial injury in the setting of ischemia/reperfusion. The most relevant cardio-protective signaling cascades include the Reperfusion Injury Salvage Kinase (RISK) (involving the activation of PI3K/AKT/MEK/Erk) [9,10,11] and the Survivor Activating Factor Enhancement (SAFE) (triggering the JAK/STAT-3) pathways [12,13,14]. Several strategies have been investigated to reduce IRI and, among them, remote ischemic pre-conditioning (RIPC) is included [15]. The RIPC procedure, which consists in a series of brief cycles of ischemia/reperfusion (I/R) far from the heart (e.g., the arm), has been described as one of the most effective cardio-protective approaches [16,17,18].

Extracellular vesicles (EV) are small anuclear bilayered lipid membrane particles that are released by almost all cell types and are enriched in several bioactive molecules including lipids, proteins, amino acids, mRNAs and miRNAs [19,20]. EVs can be classified, according to the International Society of Extracellular Vesicles (ISEV), as small EVs (less than 100 nm in diameter) and medium–large EVs (more than 100 nm in diameter) and can be isolated from many biological fluids, including blood and plasma [21]. They have been recognized as essential mediators of intercellular communication and their role is being increasingly documented in several pathophysiological settings and in cardiovascular diseases [22,23,24]. In particular, it has been reported that both myocardial damage and reperfusion by coronary artery bypass surgery (CABG) drive changes in the mRNA and miRNA content of circulating EVs [25]. Ruf et al. [26] and Paganelli et al. [27] have also shown that serum-derived EVs, released by patients with cardiovascular disease, are enriched in A2A adenosine receptors. Moreover, it has recently been demonstrated that RIPC is able to increase the number of circulating EVs and alter their RNA signature in patients undergoing CABG [28,29]. We have recently reported that EVs collected before PCI are cardio-protective [30]. However, so far, no data on EV content and functional behavior have been reported in reperfused ACS patients undergoing RIPC or sham pre-conditioning.

It has been suggested that reperfusion boosts the ischemia damage in patients experiencing PCI [2]. The aim of this study was to investigate whether and how EVs recovered after PCI from ACS patients affect reperfusion injury. For this purpose, in vitro (HMEC-1 + H9c2 co-culture) and ex vivo (isolated rat heart) approaches were performed using EVs recovered after PCI from ACS patients, randomized to receive RIPC or sham procedures. Moreover, EV cargo was also evaluated. Particular attention was devoted to the analysis of cardio-protective, stress-related and anti-apoptotic EV gene profiling.

## 2. Results

### 2.1. Clinical Features of Patients

Of the 72 patients screened, 42 were excluded from the study, while 30 unstable angina (UA; n = 12) and non-ST elevation myocardial infarction (NSTEMI; n = 18) patients were randomly allocated (1:1) to receive the sham or RIPC procedure (Figure 1). Since patients included in this study were the same as described in D’Ascenzo et al. [30], patients’ clinical and procedural characteristics reported in Table 1 are unchanged (Table 1 modified by D’Ascenzo et al. [30]).

### 2.2. EV Characterization

EVs have been characterized using different approaches. First, we analyzed serum-derived EV-RIPC and EV-naive by transmission electron microscopy (TEM). As shown in Figure 2A, vesicles in the nano-size range were observed by TEM and no differences between EV-RIPC and EV-naive were found. In Figure 2B, NanoSight analysis is reported. Again, EV size distribution and number did not show any significant difference. The Western blot analysis, reported in Figure 2C, demonstrated the presence of the exosome markers CD63, CD29 and CD81 in EV-RIPC and EV-naive. Accordingly, the GM130 protein was only detected in a human cell lysate (Ctrl-) (Figure 2C). Serum EVs from healthy subjects served as positive controls (Ctrl+). As shown by the results of GUAVA FACS analysis, EVs from patients of the two arms of the study expressed typical leukocyte, macrophage, platelet and endothelial markers (CD11b, CD14, CD62p and CD144), but not Caveolin (Figure 2D). Again, patient-derived EVs were found enriched in Troponin (TnT) (Figure 2D) [30]. Moreover, the expression of exosome markers was further evaluated using the MACSPlex kit (Figure 2E).

### 2.3. EV-RIPC and EV-Naive Fail to Protect Cardio-Myocytes from I/R Damage

The biological action of EVs from the two different patient groups was first evaluated in rat-derived cardio-myoblasts (H9c2) exposed to an in vitro model of hypoxia/reoxygenation (H/R) (Figure 3A). As shown in Figure 3B, only EV-RIPC collected from patients after PCI were able to protect H9c2 from H/R-induced injury. To simulate the in vivo behavior of circulating EVs, co-culture experiments were performed using human microvascular endothelial (HMEC-1) and H9c2 cells. To this end, co-cultures of HMEC-1 and H9c2 cells (trans-well assay) were pre-treated with EV-RIPC or EV-naive in H/R conditions. Unexpectedly, both EV-RIPC and EV-naive failed to induce protection in vitro (Figure 3C).

Since different results were obtained in vitro, the effect of EVs was also evaluated ex vivo by using a standard ischemia/reperfusion (I/R) protocol according to the Langendorff method [31]. For this purpose, isolated rat hearts were infused with 1 × 10^9^ EVs before the I/R protocol (Figure 3D). Consistent with the co-culture results, the infarct size was similar in the EV-RIPC, EV-naive and I/R groups (Figure 3E).

### 2.4. EV-RIPC and EV-Naive Do Not Trigger Canonical Cardio-Protective Pathways

Cardio-protection is associated with the activation of well-established signaling cascades, including the RISK, the SAFE and the antiapoptotic pathways [32]. Therefore, Western blot analysis was performed. The activation of the most relevant kinases, belonging to the RISK (Erk-1/2) and theSAFE (STAT-3) pathways, were investigated. In addition, the expression of the anti-apoptotic protein Bcl-2 was assessed in vitro and ex vivo. Surprisingly, we failed to detect Erk-1/2 phosphorylation in H9c2 cells, not only upon EV-naive but also EV-RIPC treatment (Figure 4A). Likewise, the phosphorylation of STAT-3 and the expression of Bcl-2 did not significantly change between the two groups of treatments (Figure 4A). Erk-1/2 and STAT-3 phosphorylation was also evaluated in the hearts differentially treated. Again, EV-RIPC and EV-naive were unable to induce Erk-1/2 phosphorylation. Of note, similar to the results obtained in our previous study, using EV-naive recovered before PCI [30], EV-naive recovered after PCI significantly reduced Erk-1/2 phosphorylation (** *p* = 0.002 I/R vs. EV-naive) (Figure 4B). Finally, both EV-RIPC and EV-naive were unable to induce STAT-3 phosphorylation and Bcl-2 expression (Figure 4B). The failure of both EV-RIPC and EV-naive to trigger STAT-3 phosphorylation may explain the lack of cardio-protection.

### 2.5. Gene Expression Profiling of EV-Naive and EV-RIPC

EVs exert their biological effects by releasing their cargo, consisting in genetic and non-genetic materials, in the target cell [20]. Among the genetic materials, mRNA is included [20]. The transfer and transduction of this genetic material contributes to EVs’ mechanism of action. We have previously shown that DUSP6 silencing in EVs was able to prevent EV-mediated cardio-protection by blunting STAT-3 activation [30]. Herein, we evaluated the expression of the DUSP6 protein in the heart when treated with EV-RIPC and EV-naive, and we compared its expression with the phosphorylation of STAT-3 (n = 35). DUSP6 was significantly higher in the hearts from animals treated with EV-RIPC than in the I/R group (* *p* = 0.02), while no significant differences were found in the isolated hearts subjected to EV-naive compared to the I/R or EV-RIPC groups (Figure 5A). We also evaluated the activation of STAT-3 in the same hearts. Consistent with the functional results (infarct size), STAT-3 activation was similar in both groups of treatment. This further confirms that STAT-3 is relevant for EV-mediated cardio-protection [30].

Gene expression in EVs was therefore evaluated by screening 84 human cardiovascular disease mRNAs. Comparing the gene expression profiling of EV-RIPC and EV-naive (Figure 5B), we found that five genes were upregulated in EV-RIPC compared to EV-naive. In particular, alpha-1-adrenergic receptors (ADRA1A), collagen type XI alpha 1 chain (COL11A1), CAMP Responsive Element Modulator (CREM), Frizzled Related Protein (FRZB) and Metallothionein 1X (MT1X) were significantly upregulated in EV-RIPC compared to EV-naive. Consistent with the expression of DUSP6 in the hearts, an increased DUSP6 mRNA (*p* = 0.0501) was also found in EV-RIPC. This supports the notion that the DUSP6 gene can be transferred from EVs to the heart tissue and therein transduced.

## 3. Discussion

In a recent study, we have demonstrated that EVs collected from ACS patients before PCI are cardio-protective [30]. In the present study, performed on EVs recovered from the same patients after PCI, we provide the first mechanistic evidence that PCI reprograms EV cargo, which specifically impairs EV-naive-induced cardio-protection and does not improve EV-RIPC’s biological action. 

Several studies have highlighted the cardio-protective role of serum-derived EVs [30,33,34]; however, a controversial role in reducing IRI has been reported [35]. It has been demonstrated that serum-derived EVs promote the development of ACS by inducing endothelial injury and inflammation [36,37]. In this regard, it has been shown that patients with cardiovascular disease release EVs enriched in A2A adenosine receptors [26,27]. Currently, preclinical studies using the RIPC procedure to induce cardio-protection have demonstrated efficacy, even using EVs from patients without ACS [38]. Conversely, the RIPC procedure in humans with cardiovascular diseases fails to induce protection [39,40]. The end-point designation, the failure to estimate long-term complications and the presence of comorbidities in humans have been proposed to explain differences between pre-clinical and clinical studies [40,41]. The lack of cardio-protection in patients who underwent RIPC procedures was further demonstrated by the recent CONDI2/ERIC-PPCI trial [42]. Accordingly, it has been recently shown that the RIPC procedure in ACS patients before PCI impairs EVs’ cardio-protective properties [30]. However, no data are so far available on the healing or harmful properties of EVs released upon PCI. Therefore, to assess if and how PCI can restrain EV activity, we herein recovered EVs from the serum of the same ACS patients, after revascularization, and evaluated their cardio-protective properties. Similar to EVs recovered before PCI [30], we did not find differences in EV size, number or cell of origin. Moreover, EVs from ACS patients, collected after PCI, expressed high levels of macrophage, monocyte, platelet and endothelial markers and, regardless of the diagnosis, were enriched in TnT.

Therefore, EVs have been used in in vitro and ex vivo IRI models and their underlying mechanism of action has been investigated. In particular, a co-culture of HMEC-1 and H9c2 was used for the in vitro experiments and the isolated rat heart for the ex vivo assay. Co-culture experiments better recapitulate the in vivo interplay between endothelial cells and cardio-myocytes and provide insight into the contribution of EV-treated endothelial cells to the functional biological response [31]. The isolated beating heart model (Langendorff model), besides containing all the organ components, provides a simple and effective method to study IRI, in which several extrinsic and intrinsic factors can be tightly controlled [43]. In particular, these models allow the performance of a global ischemia approach and a “clean” molecular analysis by removing all the effects derived from circulating cells. Our results revealed that EV-RIPC were able to protect H9c2 against H/R injury. However, since, in previous studies, we and others [30,31,44] have provided evidence that the trans-well assay is a more suitable in vitro model to predict the ex vivo EV therapeutic efficacy (potency test), EVs from the two arms of the study were also assayed in co-culture experiments. We demonstrated that both EV-RIPC and EV-naive failed to induce protection against H/R damage. Consistently, no cardio-protection was detected when the isolated rat hearts were infused with both EV-RIPC and EV-naive. These observations sustain the notion that in order to assess the therapeutic effectiveness of “cardio-protective agents”, the most reliable in vitro model should more closely recapitulate the in vivo physiological crosstalk between cardio-myocytes and endothelial cells.

The most efficient cardio-protective approaches require the activation of the RISK and/or the SAFE pathways [45]. However, it has been shown that IRI by itself [46], as the RIPC procedure, triggers the activation of Erk-1/2 [47]. Consistently, our ex vivo results demonstrated that I/R, similarly to EV-RIPC, was associated with increased Erk-1/2 phosphorylation. The possibility that Erk-1/2 activation represents a physiological and intrinsic cardio-protective mechanism has been postulated [48,49]. The activation of the SAFE cascade, alone or in combination with the RISK pathway, also contributes to cardio-protection [45]. We have previously shown that EV-naive, recovered before PCI from the same ACS patients, reduced Erk-1/2 activation, while they induced cardio-protection by eliciting STAT-3 phosphorylation [30]. We herein demonstrated that this effect was no longer established when EV-naive, collected after PCI from the same ACS patients, were infused in the isolated hearts. In fact, although EV-naive challenge significantly decreased Erk-1/2 phosphorylation compared to I/R, we failed to detect the activation of STAT-3. Similarly, EV-RIPC failed to trigger the SAFE cascade. These findings were consistent with the failure of EVs, RIPC or naive, to induce cardio-protection. Moreover, these data further verify that the activation of the SAFE pathway, rather than the RISK pathway, is instrumental for EV-mediated cardio-protection.

EVs exert specific functions by transferring their genetic and non-genetic content to target cells [22]. EV cargo depends on their cell of origin and on the micro-environmental cues of the releasing cells [50]. In particular, in endothelial cells exposed to inflammatory stimuli [31], stress signals were found to impair the EV healing properties by modifying their protein and miRNA content [51,52]. ACS per se can be considered a stress condition, further boosted by ischemic events induced by the RIPC procedure, and more importantly by PCI. This possibility is sustained by the change in gene expression profiling in EVs. Indeed, not only are EV-naive no more enriched in DUSP6 [30], but EV-RIPC rearrange their gene content and become enriched in some stress- and cell-cycle-related genes [53,54,55,56,57]. We can speculate that DUSP6 enrichment in the hearts treated with EV-RIPC may represent a challenge to counteract the increase in Erk-1/2 activation. However, DUSP6 expression is not suitable to elicit STAT-3 phosphorylation and cardio-protection. This raises the possibility that PCI, besides rearranging the mRNA profiling in EVs, impacts the vast array of protein, lipid and miRNA content in EV-naive as well, affecting their cardio-protective properties.

The most relevant limitations of this study are associated with the sample size and EV characterization. It has been extensively documented that EVs’ biological effects depend on their entire cargo, including proteins, lipids and miRNAs. In this study, only a subset of cardio-protective genes has been profiled. Therefore, further studies should be directed toward a more in-depth EV characterization in order to identify potential harmful, targetable molecules that are exploitable as therapeutics in ACS patients undergoing PCI.

## 4. Materials and Methods

### 4.1. Study Design

We extended a double-blind, randomized, placebo-controlled study (Clinical Trial number: NCT02195726). Briefly, 30 UA and NSTEMI patients were recruited from the Cardiology Department of the University of Turin from January 2019 through September 2019.

UA/NSTEMI, age >40 and <85 were the inclusion criteria, while Glomerular Filtration Rate (eGFR) <30 mL/min, previous or active cancer, body mass index (BMI) >29 kg/m^2^, diabetes mellitus, critical stenosis of the lower limbs and carotids and STEMI were the exclusion criteria. All procedures were in accordance with the principles of the Helsinki Declaration. The study protocol was approved by the local ethics committee and all participants provided written informed consent.

The RIPC protocol consisted of four 5 min cycles of manual blood pressure cuff inflation to 200 mmHg (or 50 mmHg over the baseline if systolic blood pressure was >150 mmHg) around the non-dominant arm, and this was alternated with 5 min deflations. Sham procedure was performed by inflating the cuff to 20 mmHg alternated with 5 min deflation. Based on other studies [58], EVs were collected from either radial or femoral artery blood samples after PCI (Figure 1). All data are reported as median and interquartile ranges (IQRs) ±SEM.

Patients included in the study were randomized to a different protocol, sham or RIPC (n = 15/group). After PCI procedure, three (7 mL) arterial blood samples from each patient were collected (Figure 1).

### 4.2. EV Isolation from Human Serum

After blood collection, the serum samples underwent the precipitation procedure for EV isolation. The protamine (P) (Sigma, St. Louis, MO, USA)/Polyethylene glycol (PEG 35,000; Merck KGaA, Darmstadt, Germany) precipitation solution (P/PEG; Sigma, St. Louis, MO, USA) (0.2 g PEG 35,000 and 1 mg protamine chloride/mL; 1:4) was added to the samples [30,59].

After overnight incubation at 4 °C, the mixture was centrifuged at 1500× *g* for 30 min at 22 °C and the pellet was re-suspended in sterile saline solution (NaCl 0.9%) and subjected to microfiltration with 0.22 μm filters (MF-Millipore) to remove larger vesicles [59]. 

### 4.3. EV Characterization

After isolation, different approaches were exploited for EV characterization. Firstly, EVs were counted and analyzed using a NanoSight NS300 equipped with Nanosight Tracking Analyses Analytical Software (Malvern Panalytical Ltd., Malvern, UK). For each sample, the number and size of EVs were evaluated. EV flow cytometry analysis was performed, 10126, Turin, Italy using the MACSPlex Exosome Kit (human, Miltenyi Biotec, Bergisch Gladbach, Germany), following the manufacturer’s protocol [60]. Therefore, each sample was analyzed using a CytoFLEX^®^ Flow Cytometer (Beckman Coulter, Indianapolis, IN, USA). CytExpert Software 1.0 (Beckman Coulter, Indianapolis, IN, USA) was used to analyze flow cytometric data. Moreover, EV FACS analyses were performed using GUAVA (Guava easyCyte Flow Cytometer, Millipore, Germany) [61]. EVs were detected using flow cytometry beads (Aldehyde/Sulfate latex 4% *w*/*v* 4 µm, Invitrogen, Carlsbad, CA, USA) and PE- and FITC-conjugated antibodies directed to CD11b, CD14, CD62p, CD144, Caveolin and Troponin (Dako Denmark A/S, Copenhagen, Denmark). FITC and PE mouse non-immune Isotypic IgG (Beckton Dickinson, Franklin Lakes, NJ, USA) served as controls. EVs were incubated with each antibody or isotype control antibody at 4 °C in 100 μL PBS containing 0.1% bovine serum albumin and then analyzed [30].

### 4.4. Transmission Electron Microscopy (TEM)

TEM was performed on EVs, which were resuspended in PBS, placed on 200-mesh nickel formvar carbon-coated grids (Electron Microscopy Science, Hatfield, PA, USA) and left to adhere for 20 min. The grids were incubated with 2.5% glutaraldehyde containing 2% sucrose, and, after washing in distilled water, EVs were processed as previously described [59] and observed with a Jeol JEM 1010 electron microscope (Jeol, Tokyo, Japan).

### 4.5. In Vitro Model

H9c2 and HMEC-1 cells were obtained from the American Type Culture Collection (ATCC; Manassas, VA, USA). HMEC-1 cells were grown in MCDB131 (supplemented with 10% fetal bovine serum (FBS), 10 ng/mL of epidermal growth factor, 1 μg/mL of hydrocortisone, 2 mM glutamine and 1% (*v*/*v*) streptomycin/penicillin) at 37 °C and 5% CO_2_ [62]. H9c2 were grown at 37 °C and 5% CO_2_ in Dulbecco’s modified Eagle’s medium nutrient mixture DMEM-F12 (supplemented with 10% FBS and 1% (*v*/*v*) streptomycin/penicillin) [63]. 

In order to verify the EVs’ effect on cardiomyocytes (H9c2) subjected to H/R, cells were serum-starved (in FBS 2% depleted of EVs) for 24 h. H9c2 were pre-treated with EVs (1 × 10^4^ EV/cell) for 2 h and then subjected to 2 h of hypoxia (1% O_2_, 5% CO_2_) in the presence of EVs. Subsequently, reoxygenation was performed (21% O_2_ and 5% CO_2_) for 1 h. The same protocol was applied for the co-culture experiments [31].

At the end of the H/R protocol, cell viability was assessed using the 3-(4,5-Dimethylthiazol-2-yl)-2,5-diphenyltetrazolium bromide (MTT) kit as indicated by the manufacturers. In brief, after 2 h incubation at 37 °C, dimethyl sulfoxide (DMSO) was added. The plates were read at 570 nm to obtain the optical density values [31].

### 4.6. Ex Vivo Model

Male Wistar rats (4 months old, body weight 350–400 g, Envigo Laboratories Udine, Italy) were used for the ex vivo experiments. Rats received humane care in compliance with the European Directive 2010/63/EU on the protection of animals used for scientific purposes. The local “Animal Use and Care Committee” approved the animal protocol (protocol no: E669.N.OVL). Animals were housed under controlled conditions with free access to tap water and to standard rat diet.

Rats were anesthetized (Zoletil 100 mg/Kg and Xylazine 15 mg/Kg) and heparinized (800 U/100 g b.w., i.m.) and hearts were rapidly excised, placed in an ice-cold buffer solution and weighed. Isolated rat hearts were retrograde perfused with oxygenated Krebs–Henseleit buffer solution (KHS; mM, 127 NaCl, 5.1 KCl, 17.7 NaHCO_3_, 1.26 MgCl_4_, 1.5 CaCl_2_, 11 D-glucose, pH 7.4) through a cannula inserted into the ascending aorta. Hearts kept in a temperature-controlled chamber (37 °C) were perfused in constant-flow mode. 

The protocols carried out on the isolated hearts were as follows:(1)SHAM group (n = 3) served as non-ischemic hearts (2 h);(2)I/R group (n = 5): hearts were subjected to I/R protocol [31];(3)EV-naive (n = 15): hearts were pre-treated with EV-naive (1 × 10^9^/mL) for 10 min before I/R protocol [30];(4)EV-RIPC (n = 15): hearts were pre-treated with EV-RIPC (1 × 10^9^/mL) for 10 min before I/R protocol [30].

### 4.7. Infarct Size Analysis

For the infarct size analysis, the hearts were allocated on the Langendorff apparatus. The hearts underwent I/R protocol and were contextually treated or untreated as above indicated. The infarct size assessment was performed using ImageJ software on heart slices dyed with 2,3,5-Triphenyltetrazolium chloride (TTC) [43]. At the end of each experiment, the hearts were detached from the Langendorff apparatus and the left ventricles were frozen for 2 h. After freezing, the hearts were dissected into 4–5 slices of 2–3 mm that were incubated in pre-warmed TTC stain (10 mg/mL in phosphate buffer) at 37 °C for 5 min. The ImageJ analysis program was used to calculate, slice by slice, the infarct size for each heart.

### 4.8. Microarrays and Ingenuity Analysis

Based on our previous study [30], to investigate the EV mechanism of action, EV mRNA content was analyzed by using a specific Cardiovascular Disease PCR Array. To this end, six samples (3/group), were retro-transcribed with the RT2First Strand Kit, and gene expression was analyzed using the PAHS 174Z RT2 ProfilerTM Human Cardiovascular Disease PCR Array (QIAGEN, Hilden, Germany) according to the manufacturer’s protocol. The analysis was performed using GeneGlobe QIAGEN online software. Fold regulation EV-naive sample expressions with respect to the EV-RIPC group were calculated for all samples using the ∆∆Ct method. The *p* values were obtained based on a Student’s *t*-test of the replicate ^2^(-Delta CT) values for each gene in the control group and treatment groups, and *p* values less than 0.05 are indicated. Parametric, unpaired, two-sample equal variance and two-tailed distributions were used to calculate the *p* values. Predicted functional interaction network was analyzed using STRING software. Network analysis provided information on the molecular and cellular interactions of genes/proteins within the network.

### 4.9. Western Blot Analysis

The canonical cardio-protective pathways were evaluated by analyzing the expression level of the phosphorylated STAT-3 for the SAFE pathway and the phosphorylated Erk-1/2 for the RISK cascade. To this end, the hearts (LV apexes) were lysed in RIPA buffer with proteinase inhibitors. Samples were quantified by the Bradford method (50 μg protein per sample was loaded) and equal amounts of total protein extracts were separated by SDS-PAGE and electro-transferred to nitrocellulose membrane. Anti-p-tyr705 STAT-3, anti-p-Erk-1/2 (#9131 and #9102, Cell Signaling, Danvers, MA, USA), anti-Bcl-2, anti-Vinculin (05-386 Merck/Millipore, St. Louis, MO, USA) and anti-DUSP6 (ab76310 Abcam, Cambridge, UK) antibodies were used as primary antibodies, followed by incubation with appropriate HRP-conjugated secondary antibodies (BioRad, Hercules, CA, USA). Proteins were detected with Clarity Western ECL substrate (BioRad, Hercules, CA, USA) and quantified by densitometry using analytic software (BioRad, Hercules, CA, USA) [31]. Results were normalized with respect to densitometric value of anti-vinculin antibody. EVs were lysed in RIPA buffer, and 30 μg protein per sample was loaded for Western blot. Anti-CD63, anti-CD81 (ab134045, ab109201, Abcam, Cambridge, UK), anti-CD29 (Ma5-17103, Thermo Fisher, Waltham, MA, USA) and anti-GM130 (ab52649, Abcam, Cambridge, UK) antibodies served as exosomal markers and negative markers of EV, respectively.

### 4.10. Statistical Analysis

All data from the in vitro and ex vivo experiments are reported as means ± SEM. Comparisons between two groups were carried out using the Mann–Whitney test or the paired *t*-test, while comparisons between ≥3 groups were performed using one-way ANOVA followed by Tukey’s multiple comparison test. Our data passed normality and equal variance tests. The cut-off for statistical significance was set at *p* < 0.05. In vitro and ex vivo results are representative of at least 3 independent experiments. All statistical analyses were performed using Graph Pad Prism version 8.2.1 (Graph Pad Software, San Diego, CA, USA).

## 5. Conclusions

This study first investigated the functional properties of EVs released after PCI in ACS patients. We demonstrated that PCI reprograms EV cargo, impairing, in EV-naive, their cardio-protective properties. Moreover, we provide the first evidence that the RIPC procedure rearranges the mRNA EV cargo after PCI, increasing the expression of some genes involved in stress responses and in the inhibition of cell cycle progression.

## Figures and Tables

**Figure 1 ijms-22-10270-f001:**
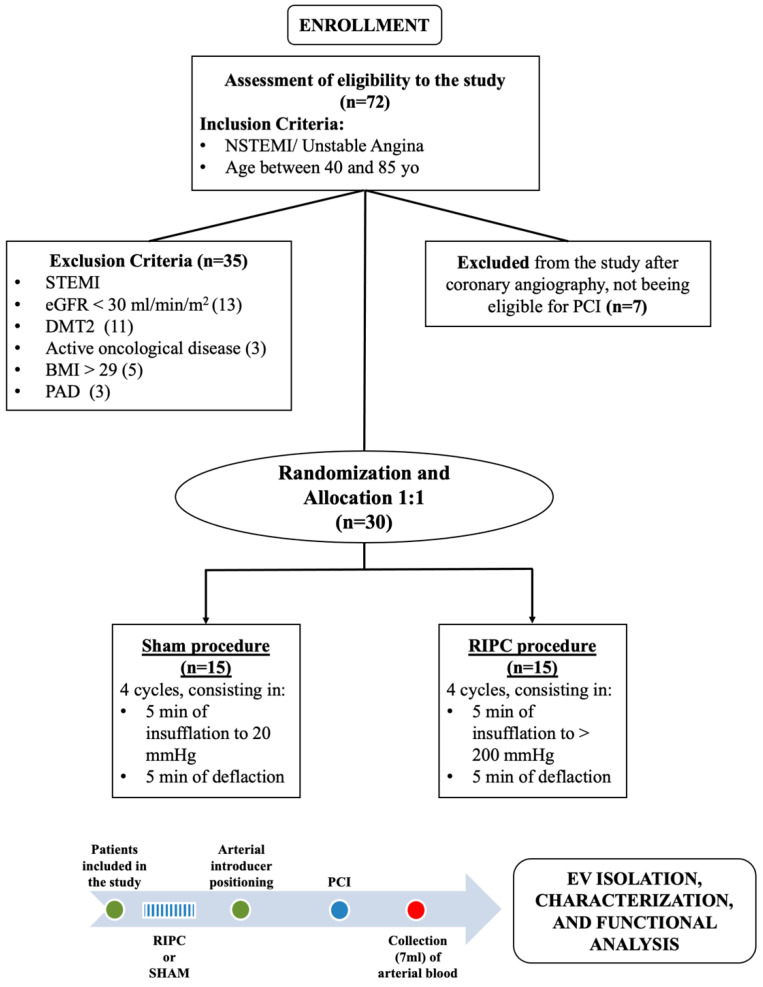
Clinical study protocol. Patients included in the trial were randomly assigned to the RIPC or sham procedure. EVs were isolated from serum samples after the PCI procedure.

**Figure 2 ijms-22-10270-f002:**
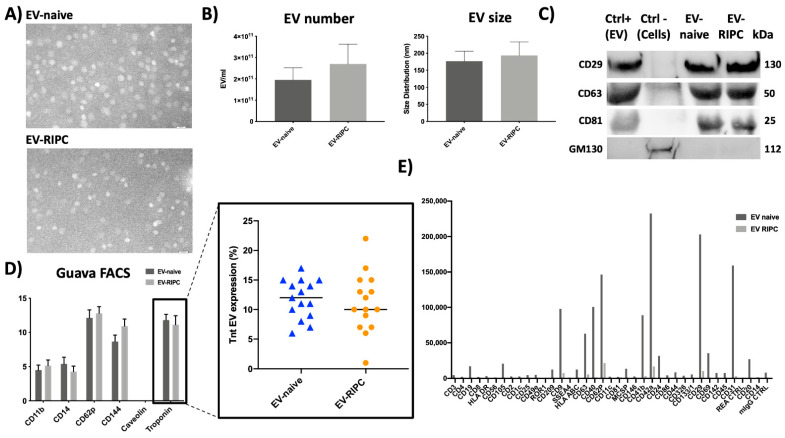
EV characterization. (**A**) Representative images of TEM performed on EV-naive and EV-RIPC (n = 3/each group). Original magnification 140 K, scale bar: 100 nm. (**B**) Histograms representing EV size and number. (**C**) Representative Western blot of exosome markers (CD29, CD63 and CD81) detected in serum-derived EVs (Ctrl+), EV-naive and EV-RIPC, and negative marker of EVs (GM130). (**D**) Flow cytometry with GUAVA FACS on EV-naive and EV-RIPC; the insert shows the EV TnT content (%) in all thirty patients. (**E**) FACS with MacsPlex kit on EV-naive and EV-RIPC.

**Figure 3 ijms-22-10270-f003:**
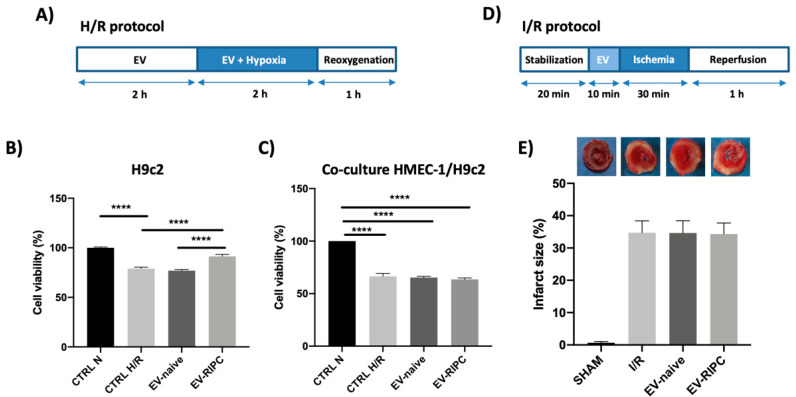
Impact of EVs in vitro and ex vivo. (**A**) Timeline of in vitro H/R protocol. H9c2 or co-culture of H9c2 and HMEC-1 were subjected to 2 h of hypoxia, followed by 1 h of reoxygenation. EVs were infused for 2 h before hypoxia and during hypoxia. (**B**) Cell viability on H9c2 cells subjected to H/R protocol, after treatment with EV-naive or EV-RIPC; data were normalized to the mean value of normoxic control. (**** *p* < 0.0001 CTRL N vs. CTRL H/R; **** *p* < 0.0001 CTRL H/R vs. EV-RIPC; **** *p* < 0.0001 EV-naive vs. EV-RIPC). (**C**) Cell viability on H9c2 co-cultured with HMEC-1 in trans-well assay; data were normalized to the mean value of normoxic control. (**D**) Timeline of ex vivo I/R protocol. The hearts were subjected to 30 min of global, normothermic ischemia, followed by 60 min of reperfusion. EVs were infused for 10 min before ischemia. (**E**) Infarct size in isolated rat hearts treated as indicated. The necrotic mass was measured at the end of reperfusion and reported as percentage of the left ventricle mass (LV; % IS/LV).

**Figure 4 ijms-22-10270-f004:**
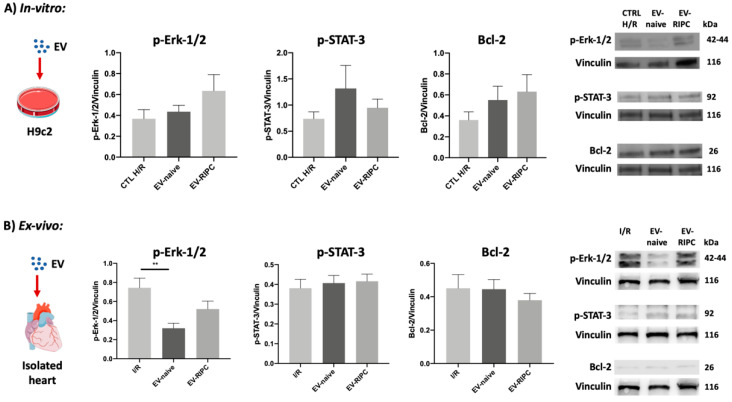
Effects of EVs on canonical cardio-protective pathways. (**A**) Representative Western blot analysis and histograms of cells subjected to H/R protocol, after EV treatment (n = 30). (**B**) Representative Western blot analysis and histograms of myocardial tissues subjected to I/R protocol, after EV treatment (n = 30). p-Erk-1/2, p-STAT-3 and Bcl-2 expression were normalized to vinculin (** *p* = 0.002).

**Figure 5 ijms-22-10270-f005:**
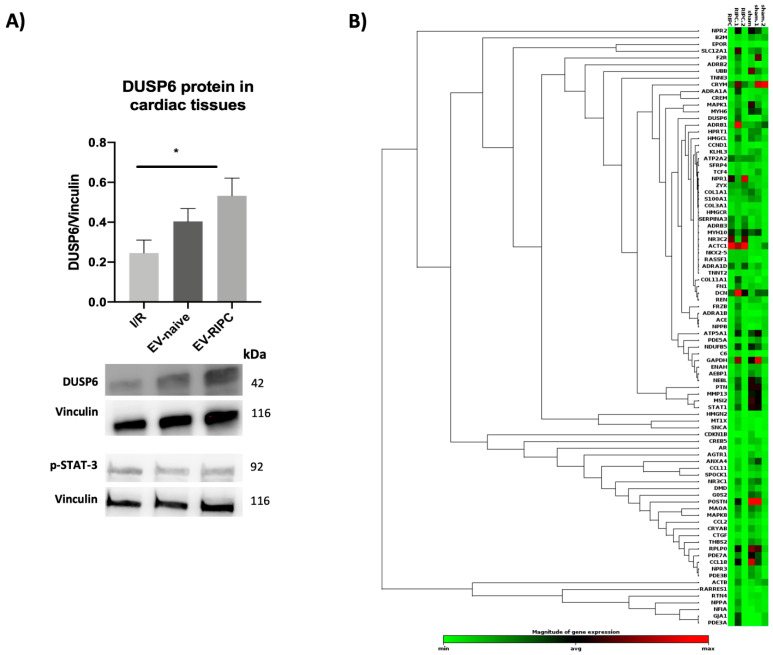
DUSP6 content in cardiac tissue and EV heatmap. (**A**) Representative images of Western blot for DUSP6 and p-STAT-3. The histogram refers to DUSP6 expression in myocardial tissue. DUSP6 and p-STAT-3 expression were normalized to vinculin (n = 30) (* *p* = 0.02). (**B**) Hierarchical clustering of the entire dataset of expressed cardiovascular-linked genes. Clustergram displaying hierarchical clustering of the entire dataset of expressed genes across two different experimental groups: EV-RIPC and EV-naive (n = 3/each group). RNAs with higher differential expression levels are represented in red, while RNAs with lower detection levels are shown in green. Genes with similar expression patterns are grouped.

**Table 1 ijms-22-10270-t001:** Patients’ clinical features at baseline.

		Sham	RIPC	*p* Value
*GENDER*	Male	12 (80%)	10 (67%)	0.68
	Female	3 (20%)	5 (33%)	
	Age	64	68	0.56
	BMI	26.6	25.4	0.23
*COMORBIDITIES*	Hypertension	11 (73%)	11 (73%)	1.00
	Smoking habit	10 (67%)	8 (53%)	0.74
	Dyslipidemia	10 (67%)	7 (47%)	0.46
	Previous AMI	7 (47%)	5 (33%)	0.71
	CKD (eGFR < 30 mL/min/m^2^)	4 (27%)	1 (7%)	0.33
	Chronic heart failure	3 (20%)	1 (7%)	0.30
	Cerebral vascular disease	2 (13%)	1 (7%)	0.50
	Previous cancer	0	3 (20%)	0.11
	COPD	0	1 (7%)	0.50
*ACS CLASSIFICATION*	NSTEMI	9 (60%)	9 (60%)	1.00
	UA	6 (40%)	6 (40%)	
*MEDICATION BEFORE*	ASA	8 (53%)	5 (33%)	0.52
*ADMISSION*	ACE-I/ARBs	7 (47%)	9 (60%)	0.69
	Beta blockers	6 (40%)	6 (40%)	1.00
	Statins	5 (33%)	4 (27%)	0.56
	Ca^2+^ channel blockers	5 (33%)	1 (7%)	0.09
	Nitrates	2 (13%)	3 (20%)	0.55
	Clopidogrel	0	2 (13%)	0.26
*CLINICAL FEATURES AT ADMISSION*	Mean eGFR (mL/min/m^2^)	75.9	80.7	0.58
	Left ventricular ejection fraction	57%	54%	0.45
	Time from onset to admission (h)	42	44	0.78
	Hemoglobin (g/dL)	14.1	13.6	0.36
*CLINICAL FEATURES DURING*	Mean number of implanted stents	2	2	1
*HOSPITALIZATION*	Mean contrast agent volume (mL)	237	228	0.77
*NUMBER OF VESSELS*	1	2 (13%)	5 (33%)	0.36
*AFFECTED BY*	2	7 (47%)	5 (33%)	
*SIGNIFICANT DISEASE*	3	6 (40%)	5 (33%)	
*COMPLICATIONS* *DURING*	New AMI during hospitalization	0	1 (7%)	0.5
*HOSPITALIZATION*	Intra stent thrombosis	0	0	
	Additional PCI	3 (20%)	2 (13%)	0.5
	Mortality during hospitalization	0	0	
BARC bleeding	0	15 (100%)	12 (80%)	0.09
	1	0	1 (7%)	
	2	0	2 (13%)	

## Data Availability

The data presented in this study are available on request from the corresponding author.

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
