# Peer review of "Percutaneous Coronary Intervention (PCI) Reprograms Circulating Extracellular Vesicles from ACS Patients Impairing Their Cardio-Protective Properties"

_ijms, 2021, doi:10.3390/ijms221910270_

Round 1
Reviewer 1 Report
This clinical trial addressed the functional properties of EV released after PCI in patients with ACS. EVs recovered after PCI from patients with ACS, randomized to receive RIPC or sham procedures, were characterized and evaluated for their biological action both in vitro (HMEC-1 + H9c2 co-culture) and ex-vivo (rat isolated heart). EV cargo was also assessed by gene expression profiling. This study shows that PCI reprograms the EV cargo, which specifically modifies the cardio-protection induced by naive EV while increasing the expression of genes involved in stress responses and in inhibiting cell cycle progression in EV-RIPC. EV-native et EV –RIPC failed to drive cardioprotection. As such this paper is important and relevant because it might be explain why remote ischaemic conditioning does not improve clinical outcomes in patients with ACS undergoing PCI (CONDI-2/ERIC-PPCI study, lancet 2019).
Although interesting and well conducted, this study is particularly difficult to follow and need majors revisions
- On the shape
- There are to much abbreviations in the abstract
- Introduction : no modification
- Then the aim of this study must be describied with attention
- Then primary endpoints ( and /or secondary endpoints)
- Then marterials and methods
- Statiscal analysis
- Then results with
- Study population (clinical features ) associated with Flow chart ( which is perfected describied)
- In vivo models and ex vivo models must be introduced and summarized in the discussion
- 2 Characterisation EV must be clarified ( one part in the methods , one part in the results)
- 3 Evaluation of I/R damage muste be clarified ( one part in the methods and one part in the results)
- With two sub-group in vivo and in vivo
- 4 Evaluation of canonical cardio-protective pathways must be clarified ( one part in the methods and one part in the results)
- With two sub-group in vivo and in vivo
- 4 Evaluation of gene expression must be clarified ( one part in the methods and one part in the results)
- With two sub-group in vivo and in vivo
g there is not legend for central figure? What is ADRA1, COL1 etc..
- Discussion ( no modification ) but in vivo models and ex vivo models must be introduced and summarized in the discussion (see below)
- There is no conclusion ++++
On form this article must be rewritten in its entirety
- in substance
- abstract The choice of DusP6 mRNA is not explained. I do not understand the last sentence; what do you mean exactly?
- There is not legend for central figure? What is ADRA1, COL1 etc…
- Introduction: it was shown that in CAD patients EV also contains A2A adenosine receptors not only mRNA and small molecules.
(Ruf j et al J cell Mol Biol 2019, Paganelli F et al Cardiovasc Res 2021). This is an important data because adenosine receptors participate into myocardial ischemia). This should be advocated. DUSP6 and genes related to stress and cell cycle should be further introduced. Please specify your choice in the introduction
Discussion : gene expression profiling is vague. It is indicated the possibility that PCI also has an impact on the wide range of proteins, lipids and miRNAs in EV-naive, by changing their cardio-protective properties. The example of the A2A adenosine receptor should be mentioned as a candidate because of its presence in plasma EV and its major role in patients with ACS. (Ruf j et al J cell Mol Biol 2019, Paganelli F et al Cardiovasc Res 2021). The last sentence of the discussion is wrong. The authors did not uevaluate all the gene implicated in the stress response. I suggest to replace « by some genes implicated… »
On the merits this article must be rewritten in its entirety
Author Response
We thank the Reviewer for His/Her appreciation and positive comments
This clinical trial addressed the functional properties of EV released after PCI in patients with ACS. EVs recovered after PCI from patients with ACS, randomized to receive RIPC or sham procedures, were characterized and evaluated for their biological action both in vitro (HMEC-1 + H9c2 co-culture) and ex-vivo (rat isolated heart). EV cargo was also assessed by gene expression profiling. This study shows that PCI reprograms the EV cargo, which specifically modifies the cardio-protection induced by naive EV while increasing the expression of genes involved in stress responses and in inhibiting cell cycle progression in EV-RIPC. EV-native et EV –RIPC failed to drive cardioprotection. As such this paper is important and relevant because it might be explain why remote ischaemic conditioning does not improve clinical outcomes in patients with ACS undergoing PCI (CONDI-2/ERIC-PPCI study, lancet 2019).
- Although interesting and well conducted, this study is particularly difficult to follow and need majors revisions
- There are to much abbreviations in the abstract
As kindly requested by the Reviewer some abbreviations have been removed in the Abstract.
- Introduction: no modification
- Then the aim of this study must be described with attention
As kindly requested, the aim of the study has been described in more detail (page 3, lines 83-93).
- Then primary endpoints ( and /or secondary endpoints)
- Then marterials and methods
- Statiscal analysis
- Then results with
- Study population (clinical features ) associated with Flow chart ( which is perfected described)
We thank the Reviewer.
- In vivo models and ex vivo models must be introduced and summarized in the discussion
As kindly requested by the Reviewer, in vivo and ex vivo models have been introduced and summarized in the Discussion (page 10, lines 254-262).
- Characterisation EV must be clarified ( one part in the methods , one part in the results)
As kindly requested by the Reviewer, EV characterization has been clarified both in Method and in Result section (page 6, line 112 and page 12, line 347).
- Evaluation of I/R damage must be clarified ( one part in the methods and one part in the results)
- With two sub-group in vivo and in vivo
As kindly requested by the Reviewer, I/R damage assessment has been clarified both in the Method and in the Result section (page 7, lines 147-148 and page 13, line 406-407).
- Evaluation of canonical cardio-protective pathways must be clarified ( one part in the methods and one part in the results)
- With two sub-group in vivo and in vivo
As kindly requested by the Reviewer, evaluation of canonical cardio-protective pathways has been clarified both in Methods and in Results (page 7-8, lines 167-171 and page 14, lines 430-432).
- Evaluation of gene expression must be clarified ( one part in the methods and one part in the results)
- With two sub-group in vivo and in vivo
As kindly requested by the Reviewer, evaluation of gene expression has been clarified both in Methods and in Results (page 8, lines 192-195 and page 13, lines 415-417).
- Discussion ( no modification ) but in vivo models and ex vivo models must be introduced and summarized in the discussion (see below)
- As kindly requested by the Reviewer, in vivo and ex vivo models have been introduced and summarized in the Discussion section (page 10, lines 254-262).
- There is no conclusion.
- As kindly requested by the Reviewer, conclusions have been included (page 14).
On form this article must be rewritten in its entirety
in substance
- abstract The choice of DusP6 mRNA is not explained. I do not understand the last sentence; what do you mean exactly?
As kindly requested by the Reviewer, this suggestion has been included in the Abstract.
- There is not legend for central figure? What is ADRA1, COL1 etc…
The legend for central figure has been added.
- Introduction: it was shown that in CAD patients EV also contains A2A adenosine receptors not only mRNA and small molecules. (Ruf j et al J cell Mol Biol 2019, Paganelli F et al Cardiovasc Res 2021). This is an important data because adenosine receptors participate into myocardial ischemia). This should be advocated. DUSP6 and genes related to stress and cell cycle should be further introduced. Please specify your choice in the introduction.
As kindly requested by the Reviewer, these two references have been included in the Introduction (page 3, lines 75-77).
As kindly requested by the Reviewer The choice of Dusp6 has been explained and included in the Introduction section (page 3, lines 91-93).
- Discussion: gene expression profiling is vague. It is indicated the possibility that PCI also has an impact on the wide range of proteins, lipids and miRNAs in EV-naive, by changing their cardio-protective properties. The example of the A2A adenosine receptor should be mentioned as a candidate because of its presence in plasma EV and its major role in patients with ACS. (Ruf j et al J cell Mol Biol 2019, Paganelli F et al Cardiovasc Res 2021). The last sentence of the discussion is wrong. The authors did not uevaluate all the gene implicated in the stress response. I suggest to replace « by some genes implicated…»
We thank the Reviewer for the helpful comments. The references have been included in the Discussion (page 10, lines 235-236).
As requested, the sentences regarding stress-genes have been modified in the Abstract, in the Discussion and in the Conclusions.
Reviewer 2 Report
The article is well-written and comprehensive. The study is correctly designed, the results are clearly and transparently presented and the set goals correspond to the conclusions. Even though this study included a relatively small number of patients from a single center, I consider that the findings are interesting and that the results obtained can make significant contributions to further large studies. I suggest emphasizing the limitations of the study and also research priorities in this field.
Author Response
We thank the Reviewer for His/Her appreciation and positive comments
The article is well-written and comprehensive. The study is correctly designed, the results are clearly and transparently presented and the set goals correspond to the conclusions. Even though this study included a relatively small number of patients from a single center, I consider that the findings are interesting and that the results obtained can make significant contributions to further large studies. I suggest emphasizing the limitations of the study and also research priorities in this field.
As kindly requested by the Reviewer, the limitations of the study and the research priorities are included in the Discussion section (page 11, lines 305-310).